# Reconfigurable Electronic Platforms: A Top-Down Approach to Learn about Design and Integration of Electronic Systems

**DOI:** 10.3390/mi13030442

**Published:** 2022-03-15

**Authors:** Almudena Rivadeneyra, Francisco J. Romero, Michael Haider, Vijay D. Bhatt, Jose F. Salmeron, Noel Rodriguez, Diego P. Morales, Markus Becherer

**Affiliations:** 1Department of Electronics and Computer Technology, University of Granada, 18071 Granada, Spain; franromero@ugr.es (F.J.R.); noel@ugr.es (N.R.); diegopm@ugr.es (D.P.M.); 2Institute for Nanoelectronics, Technische Universität München, 80333 Munich, Germany; michael.haider@tum.de (M.H.); vijay.bhatt@tum.de (V.D.B.); jf.salmeron@tum.de (J.F.S.); markus.becherer@tum.de (M.B.)

**Keywords:** education, electronics engineering, programmable platforms, rapid prototyping, reconfigurable electronics, sensors

## Abstract

This case report presents a real example of a study which introduces the use of reconfigurable platforms in the teaching of electronics engineering to establish a bridge between theory and practice. This gap is one of the major concerns of the electronics engineering students. Different strategies, such as simulation tools or breadboard implementations, have been followed so far to make it easier for students to practice what they study in lectures. However, many students still claim to have problems when they face real practical implementations. The use of reconfigurable platforms as a teaching tool is proposed to provide the students the possibility of fast experimentation, reducing both development time and the learning curve. In addition, reconfigurable platforms available on the market make this methodology suitable to be applied throughout the different courses of their curricula. The feasibility of this approach is demonstrated in a course at the M.Eng. level, where the objective is the study, design and development of electronic sensor nodes. We firmly consider, based on the students’ results and reflections collected during the course, that this methodology helps students to address the theoretical framework from a practical viewpoint, as well as to acquire some of the fundamental skills for their professional careers, such as the usage of communication protocols and embedded systems programming, in a more intuitive way when compared to traditional teaching methodologies.

## 1. Introduction

In almost any discipline, there is always a gap between theory and practical knowledge. This gap is particularly wide in some engineering fields, such as mechanical or electrical engineering [1]. This situation is especially noticeable at an educational level, where students are able to solve theoretical problems, but they are not capable of applying this background to develop practical solutions. This work describes a solution to bridge this gap in electronics at the university level, reducing the intellectual distance between the academic world and the industry. Any student of electrical and electronics engineering is expected to be able to design and implement an electronic circuit of a certain degree of complexity. To succeed in that, many different competences should be learnt during his or her study. However, apart from the mentioned gap between theory and practice, sometimes the students barely find the links and synergies among the different lectures. Therefore, students or graduates face many troubles trying to build such electronic applications, feeling insecure and wasting too much time and resources.

Basic knowledge and specific skills about sensors and actuators, signal conditioning stages, communication to other entities and storage of information are essential to design and implement any fully functional electronic system. For this purpose, conventional electronic laboratories make use of protoboards, cables, through-holes components and, sometimes, they also employ stripboard and solder the components. However, because of this lack of connection among competences and, especially, between theory and practice, students and even graduates find these kinds of tasks to be hard. Moreover, these practices are usually far from real industrial environments where, for instance, integrated circuits and surface-mounted devices (SMD) are employed.

This work proposes another type of solution to tackle the same tasks: reconfigurable electronics. This term refers to those electronics systems that can change their inner physical structure in order to cope with different incoming signals. Reconfigurable circuits are increasingly present in a wide range of applications, from consumer electronics to medicine or defense. These kinds of circuits offer a performance comparable to that obtained using application-specific integrated circuits (ASICs), with the additional advantage that they do not require neither large nonrecurring engineering (NRE) costs nor long development times [2]. These tools, apart from also being utilized at the industrial level, offer a complete framework for students to easily and quickly learn the design of electronic circuits, providing the desirable link between theory and practice.

Furthermore, students are usually encouraged to simulate their circuit designs before attending labs session, which allows them to validate their designs prior to their assembly. However, if they address the design of an analog filter or an amplifier, the number of possible configurations together with the calculus of the associated capacitors and resistors limit the possibility of testing them in practical implementations during the lecture. Reconfigurable electronics helps to tackle this issue from a top-down approach. Thanks to their predefined electronics blocks and their inherent reconfigurability, students are able to test different configurations of gain, bandwidth and/or cut-off frequency without the necessity of building different electronic circuits. Thus, once the required parameters for the designs are obtained, the assembly of a custom circuit based on discrete components becomes both easier and faster. Moreover, reconfigurable electronics platforms constitute low-priced and complete solutions for remote and online learning, providing for both students and teachers the freedom to work and learn at home. This aspect is of crucial interest for self-learning, serving as a valuable tool for people who study and work at the same time. In extreme situations, such as global pandemics, reconfigurable electronics platforms can be a link to practical laboratory competences during confinement and shutdown periods.

This paper is structured as follows. First, the different reconfigurable electronic platforms available on the market are described. After that, the competences and tools that students might learn by using these kinds of solutions are presented, followed by a real example executed in a university laboratory for Master students. Finally, the main conclusions are drawn.

## 2. Reconfigurable Electronics Platforms

Different reconfigurable solutions can be found on the market. This section describes the most common platforms, highlighting their resources, whereas a brief comparison is presented in Table 1.

### 2.1. Field Programmable Gate Arrays (FPGAs)

FPGAs consist of a matrix of logic blocks (LB) and an interconnection network. The functionality of LBs and the configuration of the interconnection network can be modified through the download in the FPGA of a set of bits, which defines the hardware configuration [3]. The LBs are typically organized in a regular matrix, which is surrounded by the interconnection network. The LBs in the periphery are connected to the input/output blocks (IOB) to enable the FPGA to communicate with an external device. FPGA are normally used to implement applications with an embedded microprocessor [4] and/or applications with dedicated processing [5,6].

### 2.2. Field Programmable Analog Arrays (FPAAs)

More recently, FPAAs have been introduced as an alternative for the rapid prototyping of analog circuits [7]. Just like FPGAs, FPAAs consist of an interconnection network that allows to route analog signals towards their different configurable analog blocks (CABs) [8]. Examples of analog functions implemented by means of CABs are filters, integrators, amplifiers or comparators, among others. In this way, many researchers have considered the use of FPAAs for their electronic designs in order to implement analog functions [9], or process analog signals [10,11,12]. Thus, it is expected that, as the FPAAs become more efficient and cheaper, they will see a growing interest for use in engineering projects, especially in those where real-time processing is essential since they receive, process and transmit signals without the need of A/D or D/A conversions [13].

### 2.3. Software-Defined Radio (SDR)

Since its emergence, the software-defined radio platforms have also attracted an increasing interest for the rapid prototyping in wireless networking, enabling a shift from the custom inflexible hardware radio platforms to the parametrizable and software reprogrammable architectures [14]. Thus, because of their inherent advantages, such as rapid design cycles, flexible real-time operations, reusable hardware, as well as ease of manufacturing and upgrading, SDRs have become an invaluable tool not only in the field research and development, but also at the educational level [15]. In fact, SDRs are already being used in several courses, contributing to a decrease in the learning curve associated with the development of communication systems, as it has been demonstrated by studies in different universities [16].

### 2.4. Programmable System-on-Chip (PSoC^®^)

PSoC^®^ is a family of SoCs, which combines configurable analog and digital circuitry with a microcontroller and a programmable routing and interconnecting network (see Figure 1). Contrary to SDRs, the PSoC^®^s integrate all the reconfigurable hardware and digital domains in the same silicon die; besides, they offer more versatility for the implementation of electronics devices, since they are not limited to radio-based applications. These platforms allow both digital and analog processing, but also they have dedicated communication interfaces such as I2C, USB, CAN, JTAG, etc. Due to their versatility, in recent years these devices have attracted a lot of interest [17,18,19,20], and can also be found in many commercial products [21].

## 3. Reconfigurable Electronics as a Teaching Tool

Taking advantage of the reconfigurability of the described commercial solutions, the designers (students, graduates, professors) have multiple available resources that can be configured as desired for each application/project. This feature is of special interest in lab courses at the M.Eng. level since it reduces the time intended for the assembly of the circuits using discrete components (which is a competence that students are expected to possess at this level). In this way, the reconfigurable electronics help to seize the lab sessions to understand and develop what the session is intended for: communication protocols, signal conditioning, A/D or D/A conversions, or measurement equipment skills.

This reason has led many professors to adopt reconfigurable electronics for their courses. For instance, Mahmoudi et al. [22] use a reconfigurable prototyping platform based on digital signal processors (DSP) to implement a wide range of power conversion circuits and systems. Given that power electronic converters have a fairly standardized composition, the use of this platform allowed them to implement and study different power conversion topologies with different control algorithms on the same board. Pimentel et al., for their part, opt for the use of a SDR for their courses on wireless communications and networks thanks to the versatility that this tool offers for implementing different modulation schemes and protocols on the same device [23]. In the same way, Morales et al. also make use of a reconfigurable platform for the development of the subject “Circuits and Electronic systems for biomedical applications” for electronics engineering students [24]. Specifically, they use the programmable analog front-ends (AFEs) of the PSoC^®^ devices to acquire different bio-signals, such as the electrocardiogram, and the integrated processing engine for their subsequent processing.

Therefore, there are several specific competences and skills that can be learned and trained by using such platforms, as follows:-*Signal Conditioning*. The use of any electronic device requires the understanding of the specifications collected in its datasheet (e.g., operating voltage, work frequency, etc.). This information is fundamental for choosing the electronic components and designing the final system. For instance, any sensor or actuator requires a signal conditioning stage that needs to match its specifications and the ones of the final system.-*Programming*. Almost any electronic application has a chip or microcontroller that needs to be programmed [25], and the same applies to these kinds of platforms. In this way, these tools are also useful to work with different programming languages (e.g., C is used to program the ARM microcontroller of the PSoC^®^, while FPGAs are usually programmed in either VHDL or Verilog).-*Communications protocols*. There are several standard wire communications protocols that are widely used to interconnect devices and modules, such as I2C or SPI [26,27]. However, some electronic chips also have their own protocols that must be implemented for their operation, not only in terms of programming but also in terms of hardware. This issue can also be addressed by means of reconfigurable electronics. Besides, SDRs also offer the possibility to implement several wireless protocols in the same hardware [28], avoiding the use of a single custom-hardware device for each wireless protocol (e.g., 82.11 b/g/n, ZigBee, BLE, LoRa, etc.).-*Connections*. The configuration of connections, i.e., which pins can be or need to be selected for each kind of function. For instance, some pins are reserved for outputs and inputs, or for the clock signal.-*Testing of final systems*. These tools provide the perfect environment to test different final system configurations without wasting too many resources, in terms of both electronic components and engineering time, giving the opportunity to have several prototypes, before creating a customized printed circuit board (PCB).

## 4. Example of Use

A lab course was designed in order to evaluate these solutions as educational tools, in which the main resource was a PSoC^®^. This section presents the formal lecture details, followed by the lecture program. Finally, both students’ opinions and professors’ experiences are summarized.

### 4.1. Lab Course Details

The official information of this lab course is found in Table 2, and in the following, the objectives, pre-requirements and expected results are detailed.
-*Objectives*. The course is based on the TUMino Sensor Node shields, which provide a unified modular framework for IoT sensors based on the Arduino UNO form factor, as shown in Figure 2. In particular, the students work on the design, fabrication and characterization of IoT sensor nodes with a main focus on power supply, physical transducers (sensors), signal conditioning, A/D conversion and PSoC^®^ programming. The students are given a set of TUMino shields, which integrate different kinds of sensors, such as temperature, humidity, air-quality or magnetic sensors, among others. In this course, the TUMino shields are used in combination with a Cypress PSoC^®^ platform. The use of this reconfigurable platform helps the students to understand in an easier way some of the basic concepts when designing an electronic device. This is possible thanks to the pre-verified “virtual chips” included in the intuitive and graphical PSoC^®^ Creator™ Integrated Design Environment (IDE) (see Figure 3). Therefore, the students can drag-and-drop the different components (analog and digital) into the design and configure them as a function of the final application requirements, thus avoiding the complexity of code-based configurations. Once the different analog and digital components are configured as desired, the students can generate the application code using the C language and upload it to the ARM^®^ Cortex^®^ microcontroller.-*Pre-requirements*. The students should have basic knowledge on electronic devices’ physics, material sciences, electronic circuits and printed circuit board (PCB) prototyping. It is suggested that the students should have participated in the modules related to energy harvesters and/or nanotechnology and/or nano-systems (or other related subjects).-*Expected results*. After completion of the module, the students should be able to evaluate sensor nodes in the framework of the so-called Internet of Things (IoT), both at the device and circuit level. They are expected to comprehensively review building blocks of IoT devices and circuits at different levels of abstraction.

### 4.2. Evaluation of the Lab Course

The success of reconfigurable electronics as a teaching tool in electronics engineering was based on questionnaires distributed at the begging and at the end of the lab course. These questionaries contributed the core of the empirical methodology of data collection together with the professors’ notes during the lab sessions and the professors’ evaluations of prototypes and reports submitted by students and the students’ presentations. All professors already had experience in similar laboratories, so that they could assess the impact of the strategy followed in this lab course.

### 4.3. Lab Course Program

The first session was intended to explain the lab course details and to plan the following sessions with the students. At the beginning of this first session, the students were asked to fill in a survey in order to get to know their previous knowledge and interest, together with their expectations for this course. The questionnaire revealed that, although all of the students knew about Arduino, only 20% of them had previously used it. None of the students had worked with neither analog nor digital sensors before. In addition, all of them acknowledged that they did not know what the PSoC^®^ was.

After that, the students were split into 2–3 people groups and asked to select one of the two proposed topics: (i) capacitive relative humidity (RH) sensors similar to the one described in [29], or (ii) resistive temperature sensors similar to the one detailed in [19]. During the next lab session, in the lab, they tested their sensors in a climatic chamber (VCL 4006 from VöTSCH) and impedance analyzer (E4990A from Keysight Technologies) to extract the calibration curve and to know the dynamic range of the sensors in the operative limits. Next, students prepared part of their project at home with the provided resources (PSoC^®^ Kit, Arduino board, sensors and some external components, such as resistors, capacitors and wires), implementing the readout circuit for the next session. They had to convert the analog sensor into an I2C digital sensor using both the analog and digital reconfigurable domains of the PSoC^®^ device. An example of the implementation is shown in Figure 3a for the readout circuit of a temperature sensor. Once the circuit was designed, all components were properly configured and the microcontroller was programmed, they again tested the sensor in the climatic chamber, connected this time to the PSoC^®^ platform and transmitting the data to the Arduino board. To do so, they had to cover the following areas: (i) circuit design, (ii) microcontroller programming, (iii) pin assignation (internally with the PSoC^®^ resources and externally between sensors and both PSoC^®^ and Arduino), (iv) communication protocol (in this case I2C, using the Arduino board as the master and the PSoC^®^ microcontroller as the slave, see Figure 3b) and (v) testing and debugging.

The last session consisted of a group presentation, where students described their designs and different tasks carried out during the course. Students also showed the functioning of their prototypes. After the presentation, both the professors and the audience asked questions related to their work.

### 4.4. Students’ Outcomes and Opinions on the Lab Course Program

Right after the defense of their projects, the students were asked, by means of an anonymous quiz, some questions related to their experience with this course. Below, the questions and a summary of their answers are presented.
*What do you like the most about this course*?

For this question, 80% of students agreed that this lab course gave them valuable knowledge and that they learned important tools for an electronics engineer, from circuit design to testing real sensor nodes. Furthermore, 20% of the students commented that this was the first time that they developed a complete system from design to implementation and characterization.
*What would you change in this lab*?

All students agreed that the course length was too short to succeed in all the tasks, especially because their previous experience in some areas was too weak and they needed to invest more effort and time than expected.
*What did you find more difficult in this lab*?

There was a general consensus that the students found the organization of the different tasks difficult, since this course required continuous collaborative work. With respect to technical difficulties, three aspects were pointed out as being especially difficult: design of conditioning circuits, microcontroller programming and configuration of the communication protocols.
*Overall satisfaction with this lab*

All students were satisfied with the course since they considered that the skills developed during the lab would help them in their professional careers.

### 4.5. Proffesors’ Experience

The professors involved in this course followed the students’ progress in the lab sessions, during mentoring hours and in the final defense. It was clear that all students were overwhelmed during the first sessions, mainly because it was their first time facing the design of a complete electronic node. We also noticed that the use of the reconfigurable platforms resulted in an increase of the required efforts in these first sessions with respect to other electronic lab courses, given that they had to go through new hardware and software tools. However, once the students became familiar with the different tools, we noticed the positive effects of this approach in their learning curve: the students not only achieved the different milestones faster than when this lecture was given in the traditional way, but they were also able to identify and fix the different bugs more easily. This latter point was possibly the greatest advantage provided by the reconfigurable platforms in this lab course, since handling these issues on the breadboard implementations built in the previous editions was really a great bottleneck for most of the students. It is worth mentioning that all professors and teaching staff involved in the course had previous experience with these types of experimental courses and lab practices, and all stated that the experience with the PSoC^®^ platform was gratifying.

## 5. Conclusions

This case report addressed the potential of reconfigurable electronics as a teaching tool. As it has been described, the inherent properties of reconfigurable devices make their use very interesting to provide the engineering students with the possibility to put into practice their theoretical knowledge in a fast and easier way. This is possible thanks to their versatility, hardware reconfigurability and short development time when compared with the traditional off-the-shelf-based designs. After describing the advantages that this technology brings to the acquisition of the basic competencies of an electronics engineering student, we presented a real case study, in which reconfigurable electronics were used to facilitate students’ understating of the different sub-areas covered in their curricula. Owing to its success as demonstrated by the students’ final defenses, survey questionnaires and the great reception by the students, with this work, we encourage other lecturers to include these kinds of methodologies in their study program.

## Figures and Tables

**Figure 1 micromachines-13-00442-f001:**
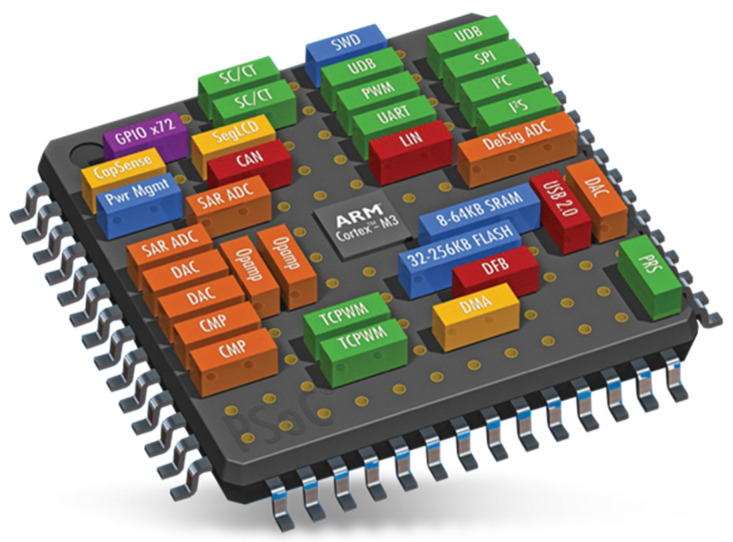
PSoC^®^ 5LP ARM^®^ Cortex™ Low-power, ARM^®^ Cortex-M3-based programmable system from Infineon Technologies. Source: https://www.mouser.co.uk/new/cypress-semiconductor/cypress-psoc-5lp-socs/ (accessed date: 8 March 2022).

**Figure 2 micromachines-13-00442-f002:**
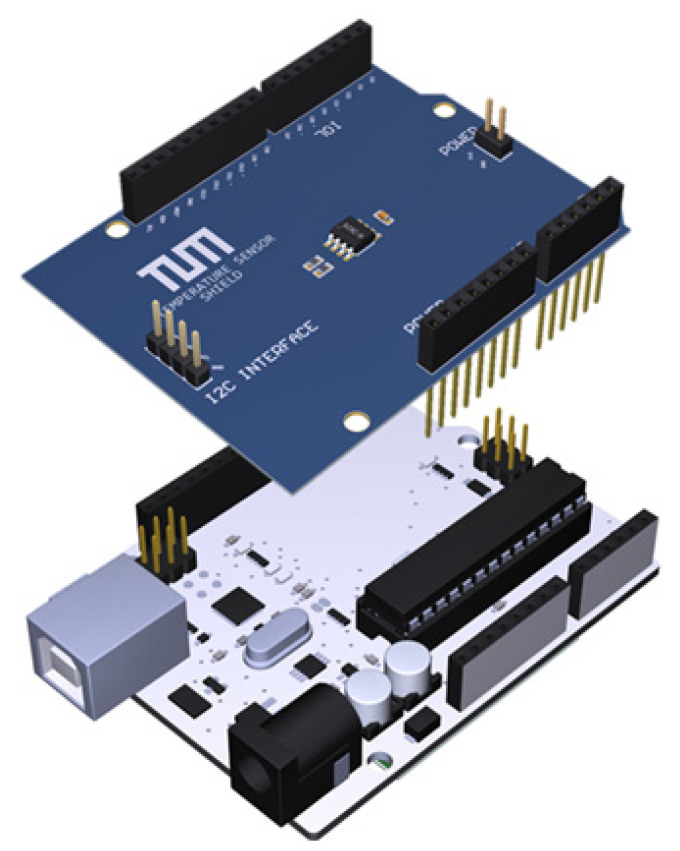
Illustration of the temperature sensor TUMino shield together with an Arduino UNO.

**Figure 3 micromachines-13-00442-f003:**
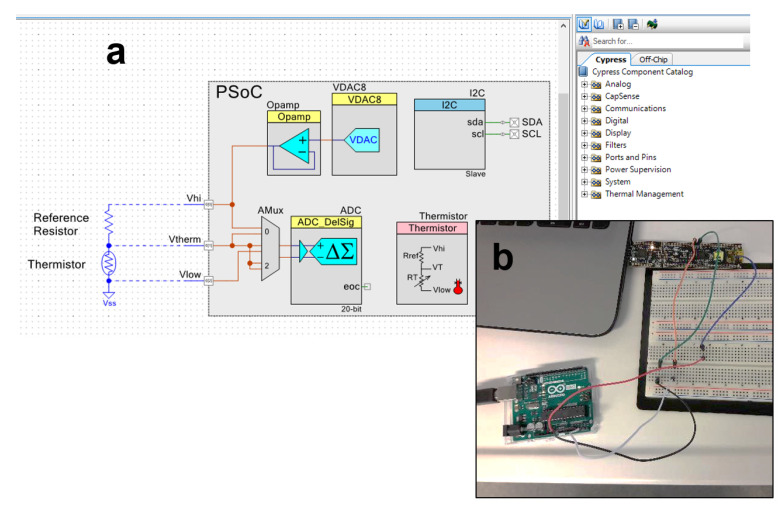
(**a**) Example of a PSoC^®^ Creator™ IDE design for a thermistor application. (**b**) Testing the I2C connection between Arduino UNO (master) and the PSoC^®^ platform (slave).

**Table 1 micromachines-13-00442-t001:** Comparison among reconfigurable electronics platforms.

Platform	Sensor Integration	Applications	Ease of Use	Cost
FPGA	Analog and digital (for digital sensors they require the implementation of the communication interface/peripheral)	Wide range of digital applications	High	High
FPAA	Analog sensors. An external microcontroller with an ADC is required to process the signals in the digital domain	Limited to analog processing, such as the front-end of analog sensors	Low	Med
SDR	N/A	Oriented to radio-based applications	Medium	Low
PSoC^®^	Easy integration of both analog and digital sensors (already includes the most common digital interfaces)	Wide range of both digital and analog applications	Medium	Low

**Table 2 micromachines-13-00442-t002:** Official information of the lecture.

Lecture Name	Sensor Node Laboratory
Lecture Type	Practice
Hours/semester	75
Line/Level	M.Eng. in Power Electronics
Registered students	20
Involved professors	5

## Data Availability

Data are available upon request to the authors.

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
