# Peer review of "Reconfigurable Electronic Platforms: A Top-Down Approach to Learn about Design and Integration of Electronic Systems"

_micromachines, 2022, doi:10.3390/mi13030442_

Round 1

Reviewer 1 Report

The paper titled ‘Reconfigurable Electronic Platforms: a top-down approach to 2 learn about design and integration of electronic systems’ presents a real example of a study which introduces the use of reconfigurable platforms in the teaching of electronics engineering to establish a bridge between theory and practice. The gap between the theoretical models and technologies really bring numerous confuses to the students. Though the simulation tools have been utilized to make it easier for students to practice what they study in lectures, many students still 16 claim to have problems when they face real practical implementations. The use of reconfigurable platforms as a teaching tool is proposed to provide the students the possibility of fast experimentation, reducing both development time and the learning curve. For this reviewer, the methodology presented in this paper will help students to address the theoretical framework from a practical viewpoint and strengthen some of the fundamental skills for their professional careers. The idea is interesting and the paper is well written. Thus, I recommend its publication in the current version.

Author Response

Thank you for your positive feedback.

Reviewer 2 Report

Reconfigurable circuits have shown great application prospects in terms of consumer electronics, medicine, and defense.  Compared to the traditional off-the-shelf designs, reconfigurable electronics have the advantages of high versatility, hardware reconfiguration and shorter development time.  This case report addressed the potential of reconfigurable electronics as a teaching tool of electronics engineering to allow engineering students to put into practice their theoretical knowledge in a fast and easier way.  The authors described different reconfigurable electronic platforms available on the market and then introduced the competences and tools that students might learn from these platforms.  Finaly, the authors presented a real example of a study that introduces reconfigurable platforms in the teaching of electronics engineering.  Overall, this article is timely and meaningful, and I would like to recommend its publication after minor revisions.

Comments:

#1.  In Section 2, the mentioned platforms might need to be listed in one table for easy comparison.

#2.  Some of the letters in the figures are too small to read and they need to be enlarged for better visibility.

#3.  Please check the format of the references and journal abbreviation carefully.

Author Response

Thank you for your feedback. All changes in the manuscript are highlighted in red. Below we reply to your comments:

#1.  In Section 2, the mentioned platforms might need to be listed in one table for easy comparison.

Following the reviewer’s suggestion, we added a table comparing the different platforms.

#2.  Some of the letters in the figures are too small to read and they need to be enlarged for better visibility.

Thank you for remark, we enlarged font size in figure 1

#3.  Please check the format of the references and journal abbreviation carefully.

We have carefully revised the references' format.

Reviewer 3 Report

Interesting and pragmatic (useful) topic.

It is adviced that authors make some changes:

  1. FORM - Title of a section should not start at previous page, without any text below, such as at page 5 (section 3. Reconfigurable Electronics as Teaching Tool ) and section 5. Conclusion.
  2. CONTENT - This paper is written as a case study, but still it needs to include some scientific methodology applied, to enable drawing conclusions. For example, there is no clear definition of empirical methodology - number of students attended, success data collection method (is it questionnaire, success at exams, types of questions at exams or structure of practical works documentation). Currently, this case study is too descriptive and does not provide any numerical proof of success of the applied method of reconfigurable electronics use in education. Each statement of success (there are present at multiple places in the text) schould be proved by some numerical, statistical proofs in experiment or questionnaire conduction.  

Author Response

Thank you for your remarks. All changes in the manuscript are highlighted in red. Below we reply to your comments:

  1. FORM - Title of a section should not start at previous page, without any text below, such as at page 5 (section 3. Reconfigurable Electronics as Teaching Tool ) and section 5. Conclusion.

We checked the format of the whole paper.

  1. CONTENT - This paper is written as a case study, but still it needs to include some scientific methodology applied, to enable drawing conclusions. For example, there is no clear definition of empirical methodology - number of students attended, success data collection method (is it questionnaire, success at exams, types of questions at exams or structure of practical works documentation). Currently, this case study is too descriptive and does not provide any numerical proof of success of the applied method of reconfigurable electronics use in education. Each statement of success (there are present at multiple places in the text) schould be proved by some numerical, statistical proofs in experiment or questionnaire conduction.  

We appreciate the reviewer's feedback. In that regard, we have included the size of the pool of students who were involved in the study in the new version (Table 2). Furthermore, we have included a subsection where we detailed all the mentioned information by the reviewer (4.2 Evaluation of the Lab Course). Numerical and statical values were already given in section 4.4 Students’ Outcomes and Opinions on Lab Course Program.

Round 2

Reviewer 3 Report

Authors made required minimal necessary changes, by adding the empirical research population data, methodology of research based on questionnaire and results of the questionnaire conduction (descriptive and numerical).

What was also expected is to have statistical data on students success at defenses of their work, to support the sentence in the conclusion related to this aspect. The statement in conclusion of having success at students defending their work is not supported with any statistically presented data in previous text.

Authors should be careful in making statements - they should be always supported by empirical data or references to prior research.